# Validation of a Novel Predictive Algorithm for Kidney Failure in Patients Suffering from Chronic Kidney Disease: The Prognostic Reasoning System for Chronic Kidney Disease (PROGRES-CKD)

**DOI:** 10.3390/ijerph182312649

**Published:** 2021-11-30

**Authors:** Francesco Bellocchio, Caterina Lonati, Jasmine Ion Titapiccolo, Jennifer Nadal, Heike Meiselbach, Matthias Schmid, Barbara Baerthlein, Ulrich Tschulena, Markus Schneider, Ulla T. Schultheiss, Carlo Barbieri, Christoph Moore, Sonja Steppan, Kai-Uwe Eckardt, Stefano Stuard, Luca Neri

**Affiliations:** 1Clinical & Data Intelligence Systems-Advanced Analytics, Fresenius Medical Care Deutschland GmbH, 26020 Vaiano Cremasco, Italy; Jasmine.IonTitapiccolo@fmc-ag.com (J.I.T.); luca.neri@fmc-ag.com (L.N.); 2Center for Preclinical Research, Fondazione IRCCS Ca’ Granda Ospedale Maggiore Policlinico, 20122 Milan, Italy; caterina.lonati@gmail.com; 3Department of Medical Biometry, Informatics, and Epidemiology (IMBIE), Faculty of Medicine, University of Bonn, 53113 Bonn, Germany; Jennifer.Nadal@ukbonn.de (J.N.); matthias.schmid@imbie.uni-bonn.de (M.S.); markus.schneider@extern.uk-erlangen.de (M.S.); 4Department of Nephrology and Hypertension, Friedrich-Alexander University of Erlangen-Nürnberg, 91054 Erlangen, Germany; Heike.Meiselbach@uk-erlangen.de (H.M.); kai-uwe.eckardt@charite.de (K.-U.E.); 5Medical Centre for Information and Communication Technology (MIK), University Hospital Erlangen, 91054 Erlangen, Germany; Barbara.Baerthlein@uk-erlangen.de; 6Fresenius Medical Care, Deutschland GmbH, 61352 Bad Homburg, Germany; Ulrich.Tschulena@fmc-ag.com (U.T.); carlo.barbieri@fmc-ag.com (C.B.); Christoph.Moore@fmc-ag.com (C.M.); Sonja.Steppan@fmc-ag.com (S.S.); Stefano.stuard@fmc-ag.com (S.S.); 7Institute of Genetic Epidemiology, Faculty of Medicine and Medical Center, University of Freiburg, 79085 Freiburg, Germany; ulla.schultheiss@uniklinik-freiburg.de; 8Department of Medicine IV–Nephrology and Primary Care, Faculty of Medicine and Medical Center, University of Freiburg, 79085 Freiburg, Germany; 9Department of Nephrology and Medical Intensive Care, Charité Universitätsmedizin Berlin, 10117 Berlin, Germany

**Keywords:** chronic kidney disease (CKD), end-stage kidney disease (ESKD), kidney replacement therapy (KRT), risk prediction, artificial intelligence, machine learning, naïve Bayes classifiers, precision medicine

## Abstract

Current equation-based risk stratification algorithms for kidney failure (KF) may have limited applicability in real world settings, where missing information may impede their computation for a large share of patients, hampering one from taking full advantage of the wealth of information collected in electronic health records. To overcome such limitations, we trained and validated the Prognostic Reasoning System for Chronic Kidney Disease (PROGRES-CKD), a novel algorithm predicting end-stage kidney disease (ESKD). PROGRES-CKD is a naïve Bayes classifier predicting ESKD onset within 6 and 24 months in adult, stage 3-to-5 CKD patients. PROGRES-CKD trained on 17,775 CKD patients treated in the Fresenius Medical Care (FMC) NephroCare network. The algorithm was validated in a second independent FMC cohort (*n* = 6760) and in the German Chronic Kidney Disease (GCKD) study cohort (*n* = 4058). We contrasted PROGRES-CKD accuracy against the performance of the Kidney Failure Risk Equation (KFRE). Discrimination accuracy in the validation cohorts was excellent for both short-term (stage 4–5 CKD, FMC: AUC = 0.90, 95%CI 0.88–0.91; GCKD: AUC = 0.91, 95% CI 0.86–0.97) and long-term (stage 3–5 CKD, FMC: AUC = 0.85, 95%CI 0.83–0.88; GCKD: AUC = 0.85, 95%CI 0.83–0.88) forecasting horizons. The performance of PROGRES-CKD was non-inferior to KFRE for the 24-month horizon and proved more accurate for the 6-month horizon forecast in both validation cohorts. In the real world setting captured in the FMC validation cohort, PROGRES-CKD was computable for all patients, whereas KFRE could be computed for complete cases only (i.e., 30% and 16% of the cohort in 6- and 24-month horizons). PROGRES-CKD accurately predicts KF onset among CKD patients. Contrary to equation-based scores, PROGRES-CKD extends to patients with incomplete data and allows explicit assessment of prediction robustness in case of missing values. PROGRES-CKD may efficiently assist physicians’ prognostic reasoning in real-life applications.

## 1. Introduction

Multiple behavioral and pharmacological interventions have proven effective in reducing the burden of risk factors for chronic kidney disease (CKD) progression [1,2,3,4]. Furthermore, timely transition management (i.e., vascular access creation and training) for patients needing Kidney Replacement Therapy (KRT) is associated with prolonged survival and reduced complication rates once on dialysis, while delayed referrals are associated with increased morbidity, mortality, and healthcare costs [5], as well as worse patient quality of life [6]. Therefore, early identification of high risk patients is an essential prerequisite of personalized clinical decision making [7,8,9].

Several prediction models were developed to assist physicians in forecasting CKD progression [10]. However, most of them have not been consistently implemented in clinical practice [9,11,12]. Indeed, the majority of published risk scores lack external validation [11,13,14], leading to suboptimal discrimination in external populations [12] and limited generalizability to clinical settings [11]. One prominent exception is represented by the Kidney Failure Risk Equations (KFREs) developed by Tangri and colleagues [15], which showed stable discrimination in different validation studies [16,17,18]. However, KFREs do not provide short-term forecasts, are not calculable for patients with incomplete data, and need re-calibration when applied to CKD populations with risk factor distributions departing from those of the original derivation dataset.

To overcome such limitations, we developed the Prognostic Reasoning System for Chronic Kidney Disease (PROGRES-CKD), a risk score application for adult patients suffering from CKD stages 3–5. PROGRES-CKD is based on a naïve Bayes Classifier (NBC) algorithm and it was trained on a large-multinational clinical dataset, reflecting real-world clinical practice. The application includes PROGRES-CKD-6 for 6-month forecasting and PROGRES-CKD-24 for 24-month forecasting.

In the present study, we reported the training and validation of both PROGRES-CKD-6 and PROGRES-CKD-24 in two independent samples of CKD patients: the FMC NephroCare cohort (European Clinical Database, EuCliD^®^, [19,20]) and the German Chronic Kidney Disease (GCKD) study cohort [21]. Moreover, we compared the PROGRES-CKD discrimination accuracy and suitability for clinical practice against the KFREs equations.

## 2. Materials and Methods

In reporting PROGRES-CKD training and validation studies we adhered to the Transparent reporting of a multivariable prediction model for individual prognosis or diagnosis (TRIPOD) statement [22] and to the Guidelines for Developing and Reporting Machine Learning Predictive Models in Biomedical Research [23].

### 2.1. Description of Naïve Bayes Classifiers

All PROGRES-CKD models are NBCs. NBCs are probabilistic models based on application of the Bayes’ theorem. The basic assumption of NBCs is conditional independence of predictors given the outcome. NBCs are represented through directed acyclic graphs (Figure 1). NBCs have been previously used in medical applications for diagnostic and prognostic reasoning in several therapeutic areas [24,25]. In fact, once derived and validated, NBCs generate metrics informing medical prognostic reasoning. First, they generate a risk score representing the expected incidence of a disease/event given a vector of known patient characteristics. Furthermore, NBCs can be used to generate value of information (VOI) statistics and impact metrics. VOI statistics represent the reduction in uncertainty (i.e., entropy) in the outcome variable that would be obtained had the value of missing variables been observed instead [26]. Therefore, it can be used to prioritize additional diagnostic testing or biomarker assays for patients with incomplete medical records. Third, NBCs can provide impact metrics (i.e., Normalized Likelihood (NL) [27]) for each observed variable. Impact metrics can be interpreted as the magnitude of association of different subsets of evidence on the outcome variable. 

### 2.2. PROGRES-CKD Training 

In this application of NBCs, we aimed at developing a model to predict the risk of KRT initiation within 6 and 24 months. The risk score is anchored at 0.00 = no risk at all to 1.00 = certainty of failure within the prediction horizons.

We derived model weights for the PROGRES-CKD by a data-driven algorithm, exploiting the wealth of information collected in the European Clinical Database (EuCliD^®^, Fresenius Medical Care Deutschland GmbH, Bad Homburg, Germany), a large, multinational, database of CKD patients. All nephrology clinics belonging to the Fresenius Medical Care (FMC) NephroCare network confer data collected for healthcare practice into this centralized data-repository. EuCliD^®^ is a fully codified database recording clinical, laboratory, socio-demographic, treatment and prescription data for each medical encounter [19,20]. Information is collected by healthcare professionals either manually or by means of interfaces to existing local data managing systems.

All non-dialysis dependent, stage 3–5 CKD patients receiving care in outpatient renal clinics belonging to the NephroCare network from 2017 to 2018 were screened for eligibility. We enrolled only patients who received at least one outpatient visit and one serum creatinine (s-cr) assessment. The endpoints of interest were KRT initiation within 6 and 24 months. We excluded patients dying before reaching the endpoint or before the end-of-follow-up (i.e., 6 or 24 months, depending on endpoint of interest). Overall, 22,535 subjects met the inclusion criteria. This initial dataset was randomly partitioned into 2 analytical samples: development (70%, *n* = 17,775), and validation (30%, *n* = 6760). The derivation of NBC weights was obtained with Hugin 8.5.

### 2.3. Measures

#### 2.3.1. Endpoint Definition

The primary endpoint was KRT initiation within 6 and 24 months. Outcome definition does not include episodes of dialysis treatment for acute and transient kidney derangement.

We defined patients as “lost to follow” when no additional s-cr assessments after end of follow-up date and no dialysis-dependence onset notes were present in the clinical records. 

#### 2.3.2. Input Variables

A list of all the variables included in the final model is provided in Table 1. The final model for the 6-month forecast incorporates 28 independent variables, while the model for the 24-month forecast includes 34 variables.

We assessed demographic, anthropometric, and lifestyle variables at index visit; blood biomarkers were collected and averaged over 12 months before index date (i.e., during the ascertainment period); their slope (i.e., change rate) was likewise calculated. Lifetime occurrence of comorbidities was evaluated by abstracting ICD10 codes [28] from outpatient medical records (Appendix A). Finally, etiologies of kidney disease were also noted.

#### 2.3.3. Definition of CKD Stages

GFR was estimated in adults using the 2009 CKD-EPI creatinine equation [29]. Patients are classified into one of the following GFR categories: (1) G1 normal or high, GFR: ≥90 mL/min/1.73 m^2^; (2) G2 mildly decreased, GFR: 60–89 mL/min/1.73 m^2^; (3) G3a mildly to moderately decreased, GFR: 45–59; (4) G3b moderately to severely decreased, GFR: 30–44; (5) G4 severely decreased, GFR: 15–29; (6) G5 kidney failure, GFR: <15.3. 

### 2.4. Design and Setting of PROGRES-CKD Validation Studies

For the validation study we randomly selected one visit from patients’ histories (index date) before occurrence of study endpoint. All information collected before the index data was used as an input variable for the model. Patients dying before reaching the endpoint or before the end-of-follow-up (i.e., 6 or 24 months, depending on endpoint of interest) were excluded. 

Based on this general design setting, we validated PROGRES-CKD models in two independent cohorts. 

#### 2.4.1. Study A

The first validation study was performed in the testing cohort derived from 30% partitioning of the clinical data abstracted from the FMC NephroCare cohort.

#### 2.4.2. Study B

A second analysis evaluated PROGRES-CKD performance using data from the German CKD study [21]. Briefly, the GCKD study is an ongoing prospective observational national study that recruited 5217 patients with CKD of various etiologies. The enrolment period started in July 2011 and ended in 2012. Patient recruitment and follow-up is organized through a network of academic nephrology centers collaborating with practicing nephrologists throughout Germany. The main study endpoints were mortality, decline in kidney function, and cardiovascular events. At the time of recruitment, patients were under nephrological care and showed either eGFR of 30–60 mL/min/1.73 m^2^ or overt urin protein in the presence of an eGFR > 60 mL/min/1.73 m^2^. In our validation analysis, only patients subjected to serum creatinine evaluation at baseline and followed for at least 2 years were considered.

#### 2.4.3. Study C

We conducted an impact study assessing concordance of nephrologists’ and PROGRES-CKD-24 ratings of risk. Four experts were asked to forecast KRT initiation risk for 78 CKD patients based on their demographic, anthropometric, and clinical data. These patients were randomly selected from the FMC NephroCare cohort and had complete clinical history up to 24 months after the index date. Information related to all input variables used by the model were extracted from existing clinical records. Information extracts for each patient were collected in real-world clinical practice by physicians during outpatient visits. Doctors were asked to rate KRT risk on a 10-point rating scale anchored at 1 (risk is negligible, almost no patient with these characteristics would require RRT within 2 years), 5 (about 50% of patients with these characteristics would require RRT within 2 years) and 10 (almost 100% patients with these characteristics would require RRT within 2 years). Risk ratings provided by the physicians were then compared to scores obtained from PROGRES-CKD-24 for the same patients. Comparative analysis included accuracy, sensitivity, and specificity based on score cut-off that maximized Youden’s Index. Thereafter, we investigated the potential impact of using risk scores provided by either experts or PROGRES-CKD-24 in referring patterns to intensified healthcare prevention programs aimed at delaying CKD progression. We simulated the use of risk estimates on a large, hypothetical CKD population of stage 3–5 CKD patients (*n* = 10,000), assuming an ESRD incidence within 24 months of 4.6% (i.e., *n* = 460 expected ESKD cases) and an intervention effect size of 1.5 (i.e., patients in the standard of care arm would face 50% higher risk of ESKD compared to those allocated in the intensified healthcare program). The intervention effect size was estimated based on expert opinion and several intensified intervention programs reported in diabetic and non-diabetic CKD [30,31,32].

### 2.5. Statistical Analysis 

We computed the cumulative incidence and the incidence density of KRT initiation events in the study population and their 95% confidence intervals based on the Poisson distribution.

Since PROGRES-CKD models are NBCs, no data manipulation was required to explicitly handle missing variables.

Model performance was evaluated by concordance statistic and calibration charts in the FMC NephroCare and the GCKD cohorts. Discrimination was quantified by calculating the area under the receiver operating characteristic curve (ROC AUC) [33]. An AUC >0.70 was considered acceptable. Calibration was visually inspected by plotting observed outcome incidence by quintiles of the risk score [34].

A further analysis investigated non-inferiority (defined as ΔAUC < 0.05) of both PROGRES-CKD-6 and PROGRES-CKD-24 relative to the KFREs [15] calibrated for the European population [16]. Briefly, Tangri’s models were developed using Cox proportional hazards regression methods in stage 3–5 CKD patients. In the present study, the following Tangri’s equations were used: (1) 4 Variables (4VAR), includes Age, Gender, eGFR, and Albumin-Creatinine Ratio (ACR); (2) 6 Variables (6VAR), includes Age, Gender, eGFR, ACR, Diabetes, and Hypertension. We could not apply the 8 Variables (8VAR) equation given the lack of serum bicarbonate assessments in both study cohorts. Non-inferiority was assessed by checking whether a one-sided confidence interval of the AUC remained entirely above the non-inferiority threshold (0.05). In case non-inferiority was achieved, we evaluated superiority of PROGRES-CKD compared to benchmark models; superiority was set at ΔAUC ≥ 0.05. Given the sequential nature of testing in a fixed order method approach, type I error is not inflated by multiple testing. Superiority was tested with the DeLong non-parametric approach [35]. Statistical significance was claimed at α < 0.05.

For study C, the following accuracy parameters were considered: Sensitivity, Specificity, Positive Predictive Value (PPV), and False Omission Rate (FOR). We also calculated the number needed to treat (NNT) in order to avoid 1 KRT event as the reciprocal of the absolute risk difference between the hypothetical prevention program and standard of care for all patients:
$$\mathrm{NNT}=(\#\text{patients int tr}/\left[\left(\#\text{patients int tr}*\text{PPV}\right)-\left(\left(\#\text{patients int tr}*\text{PPV}\right)/\left(\text{effect}-\text{size}\right)\right)\right]$$


Model training was performed using Hugin Explorer. All analyses for the validation study were performed with SAS 9.4^®^.

## 3. Results

### 3.1. Cohort Characteristics

Table 2 reports baseline demographic and clinical data of the whole FMC NephroCare cohort. Among 22,535 non-dialysis-dependent stage 3–5 CKD patients, 18,504 and 9407 patients had 6 and 24 months of follow-up, respectively. KRT events were 801 within 6 months (8.66 events/100 person-year) and 1817 within 24 months (9.66 events/100 person-year). On the other hand, KRT events in the validation sample (derived from 30% partitioning of the whole FMC cohort) were 248 (2.24 events/100 person-year) and 537 (9.36 events/100 person-year) within 6 and 24 months, respectively.

A second validation study was performed using data from the GCKD study. As shown in Table 2, a total of 4058 stage 3–5 CKD patients were included, of whom 3888 and 3687 subjects had 6 and 24 months of follow-up, respectively. RRT events were 11 within 6 months (0.5 events/100 person-year) and 80 (1.1 events/100 person-year) within 24 months.

Early CKD stages were predominantly represented in the GCKD study, whereas patients in stage 5 CKD were mostly enrolled in the FMC NephroCare cohort. Loss to follow-up within 6 months was 4031 (17.9%) and 170 (4.2%) participants, while loss to follow-up in 24 months was 13,128 (58.3%) and 371 (9.1%) participants in the FMC NephroCare and GCKD cohorts, respectively.

### 3.2. Model Discrimination in the Training and Validation Dataset from the FMC NephroCare Cohort

In the development dataset, AUC of PROGRES-CKD-6 was 0.88 (95%CI 0.86–0.89) in stage 4–5 patients, while AUC of PROGRES-CKD-24 was 0.86 (95%CI 0.85–0.87) in stage 3–5 patients. 

External validation was performed in an independent sample of patients treated in the FMC NephroCare cohort. Analysis indicated a good discriminative ability for both PROGRES-CKD-6 and PROGRES-CKD-24 models, with a concordance statistic of 0.90 (95%CI 0.88–0.91, stage 4–5) and 0.85 (95%CI 0.83–0.88, stage 3–5), respectively.

Calibration of predicted versus observed risk is represented in Figure 2.

### 3.3. Model Discrimination in the GCKD Cohort 

PROGRES-CKD models showed a good discrimination accuracy in the GCKD dataset (PROGRES-CKD-6, CKD stages 4–5, AUC = 0.91 (95%CI 0.86–0.97); PROGRES-CKD-24, CKD stage 3–5, AUC = 0.85 (95%CI 0.83–0.88)).

Evaluation of ratios of observed risk across quintiles of predicted risk indicated that the model best discriminated low and high-risk patients compared to those classified in the central quintile or risk score distribution (Figure 3).

### 3.4. Comparison with KFRE Performance

Table 3 shows the comparison in discrimination accuracy between PROGRES-CKD and KFREs equations. Since KFREs equations are computable only for complete information cases, patients with missing data were listwise deleted from this analysis. Given the large amount of missing information for ACR, we converted timed proteinuria assays (proteinuria g/24 h) into ACR when available. The conversion was based on a published correspondence table (Appendix A).

Based on the superiority test criteria, the discrimination accuracy of PROGRES-CKD-6 was greater than KFRE equations for short term RRT risk among stage 4–5 CKD patients (Table 3). PROGRES-CKD-24 discrimination was not inferior to that of the gold standard algorithms (Table 3).

### 3.5. Potential Impact Simulation

A potential impact study compared the risk of KRT estimated by nephrologists with those calculated by PROGRES-CKD-24 and investigated the potential incremental efficiency of using PROGRES-CKD compared to physicians’ assessments to inform referral to an intensified multidisciplinary prevention program to delay progression to ESKD. 

Table 4 reports ratings of CKD progression risks provided by either physicians or the prediction model. In the evaluation sample, 25 patients required KRT within 2 years, while 53 patients did not reach the study endpoint. PROGRES-CKD-24 had excellent discrimination within this dataset (AUC = 0.96), while experts’ ratings demonstrated good discrimination (average AUC = 0.79), with average sensitivity = 0.64 and average specificity = 0.85 at the optimal cut-off point (score > 6). Therefore, experts were less discriminative of endpoint occurrence compared to PROGRES-CKD-24 (ΔM-E = 0.17, *p* = 0.005). The correlation of physicians’ ratings with PROGRES-CKD-24 ratings was moderate (r = 0.50, *p* < 0.01); furthermore, experts showed different abilities to discriminate patients’ risk. (Table 4).

Figure 4 shows the results of our impact simulation. Based on the experts’ ratings (PPV = 17%; FOR = 2%), *n* = 1725 (17.3%) patients would be assigned to the high-risk category, while *n* = 8275 (82.8%) would be recommended to the standard care program (Figure 4, panel A). Based on the assumptions set for the simulation exercise (i.e., ESKD overall incidence without intervention: 2.3 events/100 patient-years; ESKD risk is reduced by 50% in the intensified intervention group) there would be 362 ESKD events overall. Therefore, in this scenario, physicians’ referral to the intensified program would delay 98 ESKD cases (i.e., an Overall Program Effect Size of 1.27). The number of patients needed to treat would be NNT = 18 (Figure 4, panel D). Conversely, risk stratification by PROGRES-CKD-24 (PPV = 48%; FOR = 1.2%) leads to referral of *n* = 732 (0.73%) patients to intensified intervention (Figure 4, panel B). In this case, 117 ESRD events would be prevented, i.e., an Overall Program Effect Size of 1.36. The number needed to treat would be NNT = 6 (Figure 4, panel D). Finally, under a hypothetical risk averse policy that would refer all stage 3 CKD patients to the intensified program, 153 ESRD events would be prevented with NNT = 65 (Figure 4, panel C).

## 4. Discussion

The present study reports the derivation and validation of the PROGRES-CKD algorithm in two independent cohorts of non-dialysis dependent CKD patients. Discrimination accuracy of PROGRES-CKD was excellent for both the short-term prediction horizon (6 months) and the long-term prediction horizon (24 months).

Of note is the fact that PROGRES-CKD-6 and PROGRES-CKD-24 had reproducible discrimination accuracy in both validation studies. The FMC NephroCare cohort included real-world clinical data of stage 3–5 CKD patients from 15 countries (Europe, South-America, Africa), while the GCKD study is a prospective CKD cohort study recruiting a wider range of NDD-CKD patients with moderate GFR impairment in Germany [21]. Given the substantial differences between the two cohorts in geographical area of recruitment (international vs. national), inclusion/exclusion criteria, and data collection strategies (real-world vs. pre-specified protocol), the observed consistency in discrimination and calibration corroborates the generalizability of PROGRES-CKD across different CKD subpopulations and clinical settings.

To further characterize PROGRES-CKD accuracy, we compared its discrimination performance against KFREs which were extensively validated in different CKD patient populations [11,17,18] and are routinely used in clinical practice. PROGRES-CKD was as accurate as KFREs for 24-month prediction in both validation cohorts and more accurate for 6-month forecasting in the GCKD study. Even though the two algorithms showed comparable performance in long-term prediction, the KFRE risk score could not be computed in a vast share of patients of the FMC NephroCare cohort because of missing information of key input variables (Figure 5). Conversely, PROGRES-CKD was available for all patients due to accurate handling of missing variables inherent to naïve Bayes classifiers (Figure 5) [36]. In fact, PROGRES-CKD potentially incorporates input from as many as 32 clinical parameters, yet its prediction can be computed with any subset of information. Therefore, PROGRES-CKD performance remained stable even for patients with many missing parameters representative of a real-world clinical practice setting. Furthermore, by assessment of VOI metrics, PROGRES-CKD allows the graphical representation of the uncertainty around prediction due to missing data. Given that VOI metrics are calculated for each missing clinical parameter within the patient’s health records, they can be used to rank the potential prognostic benefit of additional diagnostic testing or biomarker assays for patients with incomplete medical data. These peculiar features of PROGRES-CKD significantly increase its clinical usability in that they enable to address the problem of missing predictors in real-world data [17] by exploiting the full wealth of information collected in routine clinical practice.

One additional advantage of NBCs such as PROGRES-CKD over traditional equation-based prediction tools rest in their ability to generate personalized, patient-specific impact metrics representing the relative contribution of each predictor to a patient’s risk. Impact metrics can be used to estimate the potential impact of interventions addressing modifiable risk factors. This has important implications for patient care, since there can be considerable heterogeneity in underlying diseases, demographics, co-morbidities, and risk for progression among CKD patients and, consequently, optimal intervention strategies might deviate between patients with the same overall risk estimate depending on their individual high impact risk parameters. Therefore, both VOI and impact metrics could help physicians within their decision-making processes in tailoring interventions according to each individual patient’s needs and characteristics [37]. Adoption of a more personalized clinical approach would lead not only to improved CKD clinical management (targeted diagnostic and treatment investigations with minimum adverse events and maximum efficacy, and consequently increased adherence to treatment), but it could also contribute towards optimizing the utilization of healthcare resources. In fact, ranking clinical parameters by their impact on risk score computation helps physicians’ reasoning on priority and enables strategic and rational formulation of therapeutic plans considering both patient/disease-related factors and resource availability.

One specification of PROGRES-CKD allows the identification of patients whose kidney function is more likely to deteriorate within 6 months, a feature enabling timely referral to vascular access creation services and transition management [38,39]. The potential advantages of accurate short-term progression are two-fold. Patients starting on chronic dialysis with an arteriovenous fistula (AVF) rather than catheter have improved clinical outcomes in terms of survival, hospitalization, and complications [40]. On the other hand, inappropriate AVF creation in stage 4 and 5 patients who do not rapidly progress to KF is associated with complications and premature loss of patency [38].

Accurate risk prediction is a challenging task for physicians in real-world clinical practice, due to a number of disease, clinician, and organization related factors, including: inherent heterogeneity and variability in CKD progression rates [41,42], incomplete information, unrecognized case ambiguity, overconfidence leading to reduced analytical scrutiny, wrong perception of average population risk, over-generalization, fatigue, working overload, aging, altered affect impairing executive memory, switch of analytic scrutiny, and inexperience [43,44,45,46,47,48]. Therefore, readily available risk scores which prove to be accurate, generalizable to a wide array of CKD subpopulations and settings, and robust to missing data patterns observed in real-life applications may considerably assist clinical decision making, particularly when providing the opportunity to simulate the impact of interventions to individual patient cases.

In order to estimate the potential impact of improved prognostication around CKD progression on process outcomes, clinical outcomes, and costs [38,49], we conducted a simplified simulation using PROGRES-CKD as a patient stratification system for referral to intensified prevention programs for non-dialysis dependent (NDD)-CKD patients. In our simulation, risk estimates provided by either PROGRES-CKD or nephrology experts were used to stratify CKD patients. Subjects assigned to the “high-risk” category are referred to an intensified healthcare program aimed at reducing the risk of CKD progression. Our analysis suggested that PROGRES-CKD-driven referral to the intensified program would be more effective and largely more efficient than referral patterns determined by both healthcare expert risk assessment and an “all-in strategy” (i.e., all patients are referred to the intensified healthcare program when they reach stage 3 CKD). Therefore, personalized, risk-based referral may improve the efficiency of healthcare systems by enhancing the appropriateness of resource allocation in terms of direct expenditures and staff utilization. Personalized referral, however, is not just a matter of mere efficiency. In fact, inappropriate referral to the intensified intervention would involve unnecessary medicalization with greater risks of adverse events, impoverishment of quality of life even in people with a very low risk of progression, increased rate of therapeutic fatigue, and reduced adherence. Conversely, accurate and reliable patient stratification helps physicians and healthcare providers balance individual patient needs with overall resource utilization, ultimately leading to more effective care for both the individual patient and the population [50].

## 5. Limitations

Validation of risk score should be considered a continuous process of generalization tests rather than a single experiment. While the performance of PROGRES-CKD was stable in both well-conducted longitudinal cohort studies (i.e., GCKD) and historical cohorts of real-life practice (i.e., FMC NephroCare), evidence concerning PROGRES-CKD robustness with real-world-representing clinical practices outside FMC NephroCare is still missing. For this reason, PROGRES-CKD undergoes a periodical process of performance monitoring while external cohorts for validation exercises are actively sought for.

## 6. Conclusions

The Prognostic Reasoning System for CKD patients (PROGRES-CKD) demonstrated excellent discrimination accuracy in two independent cohorts of NDD-CKD patients. The underlying models provide accurate prediction for both 24 and 6 months KRT risk. Contrary to traditional equation-based algorithms which cannot be applied to a large proportion of patients with incomplete data, PROGRES-CKD extends to all patients and allows explicit assessment of prediction robustness in case of missing values for key risk factors. Furthermore, PROGRES-CKD enhances prognostic reasoning by providing patient-specific impact metrics representing the relative contribution of each predictor to a patient’s risk and can be used to estimate the potential impact of tailored interventions in addressing individual and modifiable risk factors. While PROGRES-CKD-24 may contribute to efficient and effective referral to intensified prevention programs for NDD-CKD patients, prediction of short-term outcomes (PROGRES-CKD-6) can be a key enabler of timely AVF creation and transition management. Given these results, both PROGRES-CKD algorithms reported here have the potential to advance current standards in routine CKD risk estimation, patient stratification, and individualizing interventions.

## Figures and Tables

**Figure 1 ijerph-18-12649-f001:**
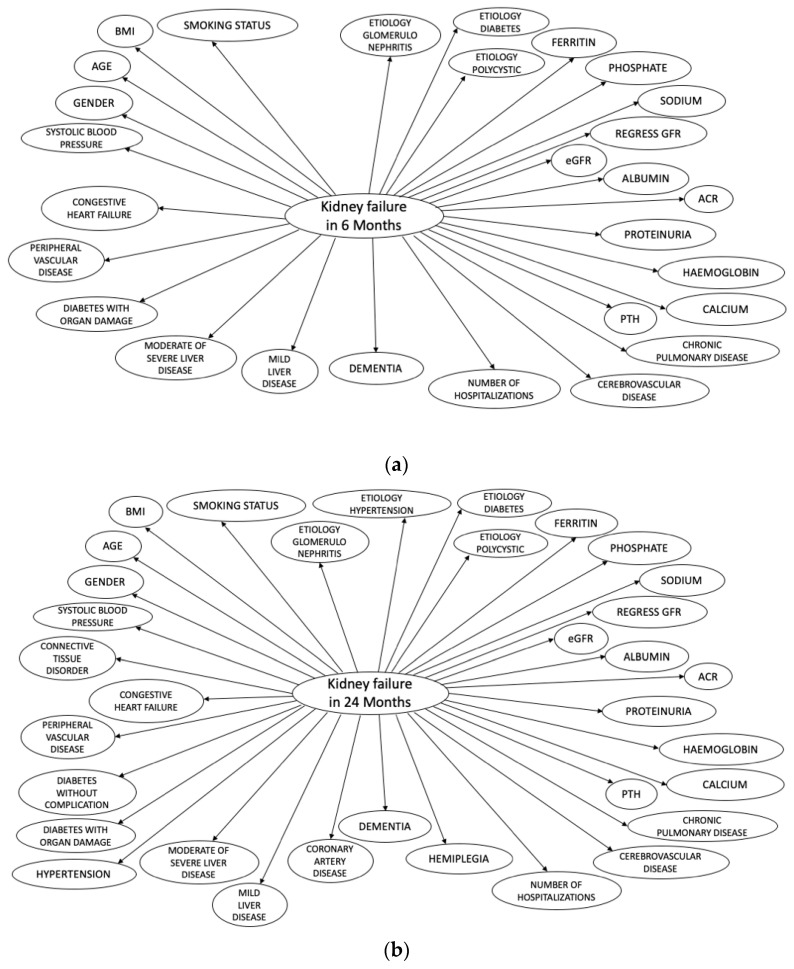
The Bayesian Network structure of PROGRES-CKD. (**a**) PROGRESS-CKD-6; (**b**) PROGRESS-CKD-24.

**Figure 2 ijerph-18-12649-f002:**
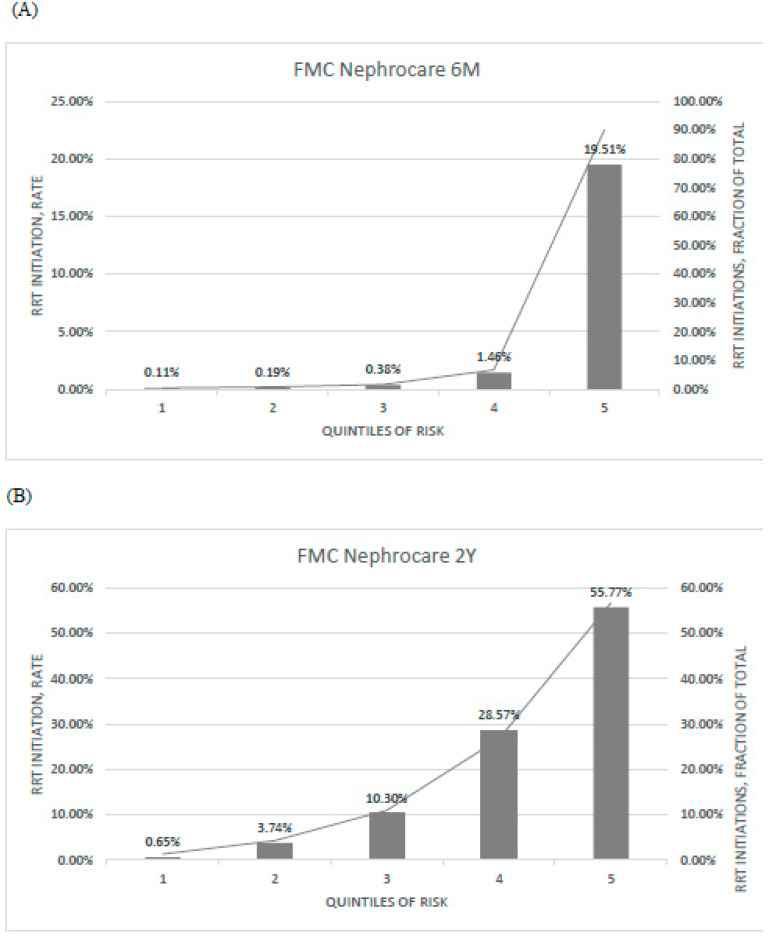
Calibration of (**A**) PROGRES-CKD-6, and (**B**) PROGRES-CKD-24 in the FMC cohort. Bar graph denotes the incidence of RRT initiation events observed in each quintile of risk (left axis); line graph denotes the fraction of RRT initiation events in each quintile with respect to the total number of RRT initiation events (right axis). Endpoint horizons: 6 months for PROGRES-CKD-6; 24 months for PROGRES-CKD-24.

**Figure 3 ijerph-18-12649-f003:**
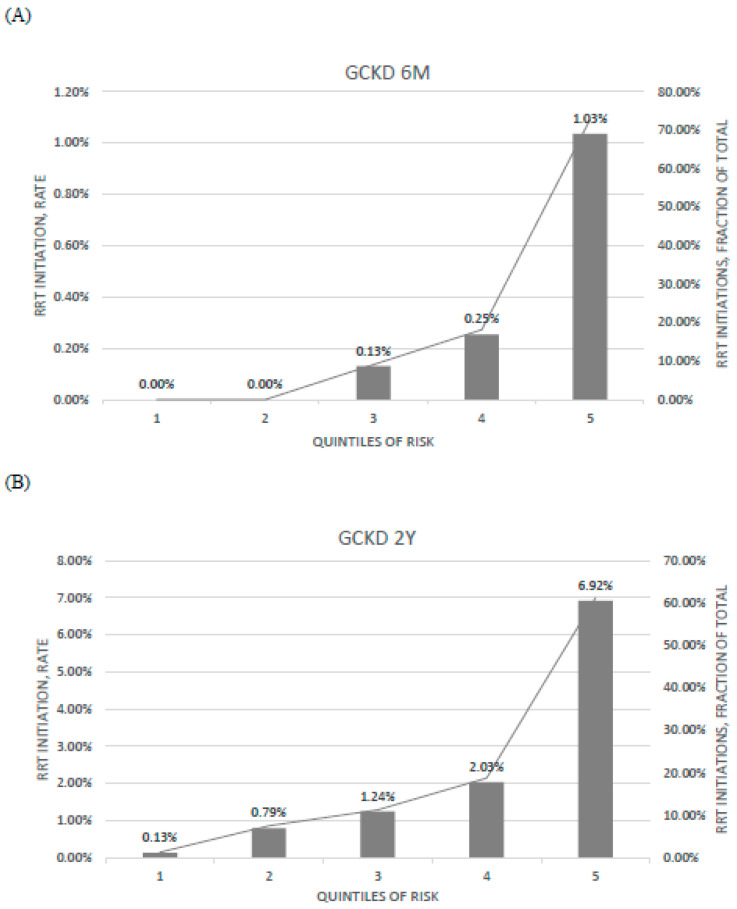
Calibration of (**A**) PROGRES-CKD-6, and (**B**) PROGRES-CKD-24 in the GCKD cohort. Bar graph denotes the incidence of RRT initiation events observed in each quintile of risk (left axis); line graph denotes the fraction of RRT initiation events in each quintile with respect to the total number of RRT initiation events (right axis). Endpoint horizons: 6 months for PROGRES-CKD-6; 24 months for PROGRES-CKD-24.

**Figure 4 ijerph-18-12649-f004:**
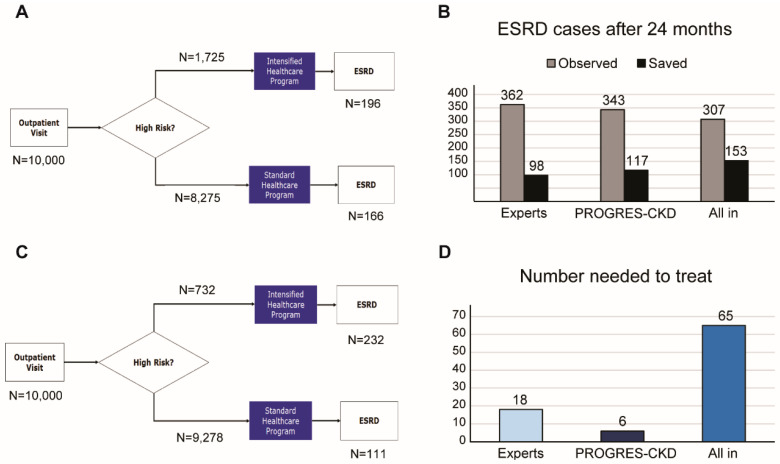
Potential impact simulation of PROGRES-CKD-24 implementation in a hypothetical CKD cohort. Flowcharts showing patients’ referral to intensified intervention programs based on (**A**) experts’ ratings, and (**B**) PROGRES-CKD scores; (**C**) Number of ESKD events within 24 months: both observed and saved cases are shown; D) Number of patients needed to treat to save 1 patient; “all-in strategy” involves referral of all stage 3 patients to the intensified healthcare program. Abbreviations: ESKD, end-stage kidney disease; NNT, Number needed to treat.

**Figure 5 ijerph-18-12649-f005:**
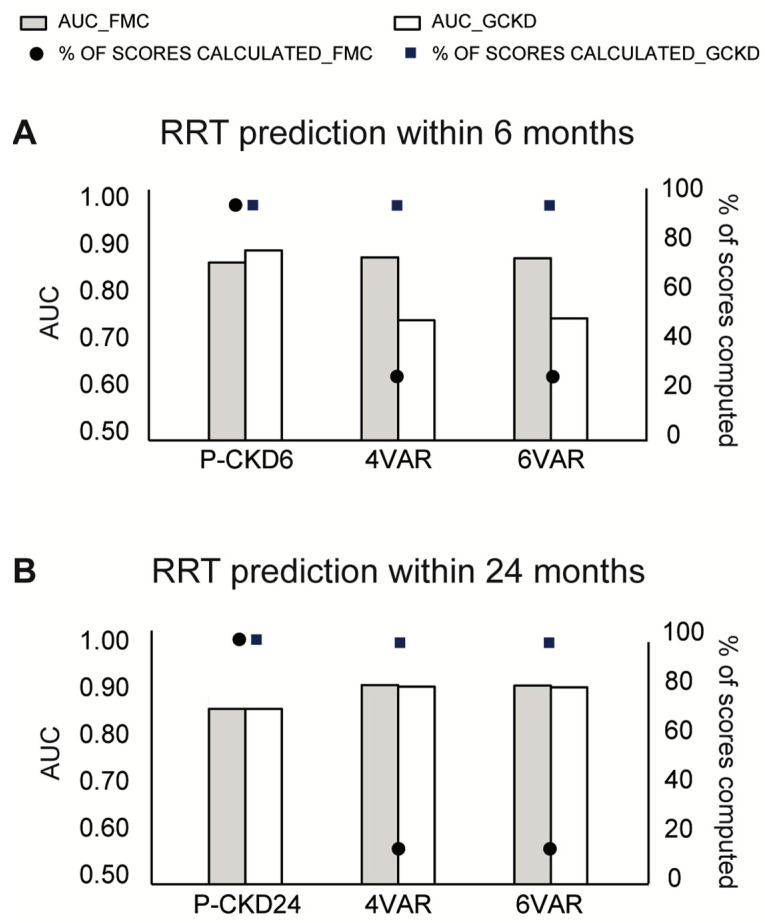
Discrimination ability of PROGRES-CKD and KFREs and percentage of computed scores by each prediction tool. Only cases with complete medical information were included in this analysis. (**A**) RRT prediction within 6 months; (**B**) RRT prediction within 24 months. Bars denote AUC (left y-axis), while dots denote the percentage of computed scores on the total number of recruited patients in each cohort (right y-axis). Abbreviations: P-CKD6, PROGRES-CKD-6; P-CKD24, PROGRES-CKD-24; 4VAR, KFRE 4 variables; 6VAR, KFRE 6 variables.

**Table 1 ijerph-18-12649-t001:** Variables included in PROGRES-CKD models.

		PROGRES-CKD-6	PROGRES-CKD-24
Group	Variable	*n* = 28	*n* = 34
Demographics and anthropometrics			
	Age	X	X
	Gender	X	X
	BMI, Kg/m^2^	X	X
	Smoking status	X	X
Kidney function			
	Albumin, g/dL	X	X
	Albumin Creatinine Ratio (ACR), mg/mmol **	X	X
	Calcium, mg/dL	X	X
	eGFR, (ml/min/173 m^2^)	X	X
	regressGFR *	X	X
	Hemoglobin, g/dL	X	X
	Phosphate, mg/dL	X	X
	Urine protein, g/24 h	X	X
	Parathyroid hormone, ng/L	X	X
	Sodium, mmol/L	X	X
	Ferritin, microg/L	X	X
Etiology of kidney disease			
	Diabetes	X	X
	Hypertension		X
	Glomerulonephritis	X	X
	Polycystic	X	X
Comorbidities			
	Cerebrovascular disease	X	X
	Chronic Pulmonary Disease	X	X
	Congestive heart failure	X	X
	Connective tissue disorder		X
	Coronary artery disease		X
	Dementia	X	X
	Diabetes with organ damage	X	X
	Diabetes without complications		X
	Hemiplegia		X
	Hypertension		X
	Mild liver disease	X	X
	Moderate or severe liver disease	X	X
	Peripheral vascular disease	X	X
Other			
	Number of hospitalizations	X	X
	Systolic blood pressure	X	X

* Slope of linear regression of eGFR values over the last 12 months. ** Urine Protein-Creatinine Ratio was converted to ACR by ACR = Urin protein*PCR (Urine protein = 0.6) (please, see the Appendix A for the conversion table).

**Table 2 ijerph-18-12649-t002:** Baseline characteristics of patients from the FMC NephroCare and GCKD cohorts.

	FMC Cohort		GCKD Cohort	
Variable	*n*	Mean ± SD or Median (IQR) or *n* (%)	*n*	Mean ± SD or Median (IQR) or *n* (%)
Stage 3	11,965	11,965 (53.1%)	3593	3593 (88.54%)
Stage 4	8026	8026 (35.62%)	460	460 (11.34%)
Stage 5	2544	2544 (11.29%)	5	5 (0.12%)
Age (year)	22,535	72.15 ± 11.7	4058	62.12 ± 10.50
BMI (kg/cm^2^)	21,655	30.63 ± 10.92	4015	30.03 ± 5.91
eGFR ((mL/min/1.73 m^2^)	22,535	31.93 ± 13.4	4058	41.92 ± 9.76
Albumin (g/dL)	19,004	4.19 ± 0.4	4055	3.85 ± 0.42
Ferritin (µg/L)	7303	222.18 ± 260.98	1044	200.48 ± 196.11
Hemoglobin (g/dL)	21,916	12.65 ± 1.83	3978	13.49 ± 1.69
Phosphate (mg/dL)	20,362	3.65 ± 0.74	4058	3.45 ± 0.64
Calcium (mg/dL)	20,686	9.36 ± 0.73	4058	9.07 ± 0.63
Sodium (mmol/L)	20,612	140.17 ± 3.16	4057	139.70 ± 3.14
PTH (ng/L)	9466	131.84 ± 150.12	0	-
ACR (mg/mmol)	90	138.67 ± 568.28	3999	393.63 ± 888.48
Proteinuria (g/24 h)	8780	3.58 ± 150.29	0	-
Systolic (mmHg)	17,963	137.33 ± 18.41	4030	140.27 ± 20.53
CRP (mg/L)	13,468	4.23 (7.63)	4056	2.41 (4.27)
Glucose (mg/dL)	19,499	126.45 ± 48.59	0	-
HDL Cholesterol (mg/dL)	7074	48.3 ± 16.74	4051	50.72 ± 17.35
LDL Cholesterol (mg/dL)	7084	107.59 ± 219.29	4051	116.33 ± 42.93
Triglyceride (mg/dL)	15,191	142.77 (95.72)	4050	173.38 (126.45)
hsTNT (ng/L)	0	-	3976	13 (11)
Uric Acid (mg/dL)	20,273	6.68 ± 1.61	4058	7.40 ± 1.92
Gender (M)	22,535	11,349 (50.36%)	4058	2510 (61.85%)
Etiology Diabetes	22,535	3614 (16.04%)	4058	666 (16.41%)
Etiology Polycystic	22,535	477 (2.12%)	4058	157 (3.87%)
Etiology Hypertension	22,535	5281 (23.43%)	4058	1011 (24.91%)
Etiology Glomerulonephrite	22,535	987 (4.38%)	4058	623 (15.35%)
Smoking status: ex-smoker	3502	3502 (15.54%)	1819	1819 (44.96%)
Smoking status: no smoker	10,066	10,066 (44.67%)	1649	1649 (40.76%)
Smoking status: smoker	2274	2274 (10.09%)	578	578 (14.29%)
Alcohol: abuse	8636	8636 (38.32%)	771	771 (19.10%)
Alcohol: moderate	0	0 (0%)	3265	3265 (80.90%)
Alcohol: abstinence	6984	6984 (30.99%)	0	0 (%)
Peripheral Vascular Disease	22,535	1875 (8.32%)	4058	424 (10.45%)
Coronary Artery Disease	22,535	4336 (19.24%)	4058	908 (22.38%)
Congestive Heart Failure	22,535	1887 (8.37%)	4058	776 (19.12%)
Cerebrovascular Disease	22,535	1876 (8.32%)	4058	472 (10.52%)
Connective Tissue Disorder	22,535	399 (1.77%)	0	-
Cancer	22,535	2469 (10.96%)	4058	532 (13.11%)
Diabetes	22,535	9021 (40.03%)	4058	1545 (38.07%)
Anemia	22,535	9800 (43.49%)	4058	1057 (26.05%)
Hypertension	22,535	17,871 (79.3%)	4058	3951 (97.36%)
Atrial Fibrillation	22,535	2337 (10.37%)	4058	876 (21.59%)
Diabetes Without Complications (CCI)	22,535	3013 (13.37%)	4058	1545 (38.07%)
Chronic Pulmonary Disease (CCI)	22,535	1618 (7.18%)	4058	285 (7.02%)
Psychiatric Disease	22,535	177 (0.79%)	0	-
Liver Disease	22,535	987 (4.38%)	0	-
RRT in 24 months	9407	1817 (19.32%)	3684	80 (2.17%)
RRT in 6 months	18,504	801 (4.33%)	3888	11 (0.28%)

**Table 3 ijerph-18-12649-t003:** Comparison between discrimination ability of (A) PROGRES-CKD-6 and (B) PROGRES-CKD-24 and that of Tangri’s Kidney Failure Risk Equations (KFREs) in the FMC and the GCKD cohort. The two scores were computed considering only complete cases (column “Effective sample size”), while patients with missing data were not included in the analysis. Endpoint horizons: 6 months for PROGRES-CKD-6; 24 months for PROGRES-CKD-24. Imputation method: Listwise. Non-inferiority was defined as AUC < 0.05, while superiority was set at ΔAUC ≥ 0.05. * Delta AUC: AUC of Tangri’s KFRE–AUC of PROGRES-CKD model.

Model	Validation Cohort	Comparator Model	AUC PROGRES-CKD	Delta AUC *	*p*-Value	Effective Sample Size
PROGRES-CKD-6						
	FMC NephroCare					
		4VAR	0.90	−0.012	0.3255	927
		6VAR	0.90	−0.016	0.2220	927
	GCKD					
		4VAR	0.91	−0.146	0.0016	459
		6VAR	0.91	−0.149	0.0013	459
PROGRES-CKD-24						
	FMC NephroCare					
		4VAR	0.87	0.020	0.0483	1081
		6VAR	0.87	0.018	0.0888	1081
	GCKD					
		4VAR	0.85	0.030	0.0105	3999
		6VAR	0.85	0.027	0.0246	3999

**Table 4 ijerph-18-12649-t004:** PROGRES-CKD-24 and Experts’ ratings of CKD progression risk.

		Experts			
	PROGRES-CKD-24	Expert 1	Expert 2	Expert 3	Expert 4
AUC	0.96	0.84	0.72	0.86	0.76
Sensitivity	0.76	0.80	0.50	0.75	0.60
Specificity	0.96	0.84	0.89	1.00	0.82

## Data Availability

We are unable to share the raw clinical data of the FMC NephroCare because data sharing would violate the terms and conditions under which Fresenius Medical Care acquired the data. Data from the GCKD study are not publicly available. External collaborators with a specific research proposal can access deidentified participant data only after review and approval of their proposal by the steering committee.

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
