# Peer review of "Validation of a Novel Predictive Algorithm for Kidney Failure in Patients Suffering from Chronic Kidney Disease: The Prognostic Reasoning System for Chronic Kidney Disease (PROGRES-CKD)"

_ijerph, 2021, doi:10.3390/ijerph182312649_

Round 1

Reviewer 1 Report

The authors trained and validated the Prognostic Reasoning System for Chronic Kidney Disease (PROGRES-CKD), a novel algorithm predicting end-stage kidney disease (ESKD) onset within 6 and 24 months in adult patients. PROGRES-CKD was trained on 17,775 stage 3-to-5 CKD patients treated in the Fresenius Medical Care (FMC) Nephrocare network. The following variables were assessed: demographic, anthropometric and life-style variables at index visit; blood biomarkers (their change rate was calculated), life- time comorbidities, and etiologies of kidney disease. The algorithm was validated in a second independent FMC cohort (N=6,760) and in the German Chronic Kidney Disease study cohort (N=4,058). In addition, accuracy of the PROGRES- CK was compared with Kidney Failure Risk Equation (KFRE). PROGRES-CKD was non-inferior to KFRE for the 2-year prediction and proved more accurate for the 6 month ESRD prediction in both validation cohorts. In the FMC validation cohort, PROGRES-CKD was computable for all patients, whereas KFE was could be computed for complete cases only (i.e. 30% and 16% of the cohort in 6- and 24-month horizons). Also, PROGRES-CKD enhances prognostic reasoning by providing patient-specific impact metrics representing the relative contribution of each predictor to patient’s risk and can be used to estimate the potential impact of tailored interventions in addressing individual and modifiable risk factors. Furthermore, risk estimates provided by either PROGRES-CKD or nephrology experts were used to stratify CKD patients. Subjects assigned to the “high-risk” category were referred to an intensified healthcare program aimed at reducing the risk of CKD progression which was more effective and largely more efficient against referral patterns determined by healthcare expert. The underlying models provide accurate risk prediction for both 24 and 6 months kidney replacement therapy. While PROGRES-CKD-24 may contribute to efficient and effective referral to intensified prevention programs for NDD-CKD patients, prediction of short-term outcomes (PROGRES-CKD-6) can be a key enabler of timely AVF creation and transition management.

Comments

The study was done on a respectable number of patients. The methodology of patient selection is correct, as is the length of follow-up. I hope that the complicated processing of the variables used will be simplified by an adequate computerized program that will enable the automated calculation of the predicted risk.

The case presented in the non-published material shows that the patient has a low risk of CKD progression during 24 months. It would be important to add (the methodology) which level of risk shows that the CKD progression will be exactly in 6 or 24 months.

The obtained results indicate that both PROGRES- CKD algorithms have the potential to advance current standards in routine CKD risk estimation, patient stratification and individualizing interventions.

Author Response

Reviewer #1

  1. The study was done on a respectable number of patients. The methodology of patient selection is correct, as is the length of follow-up. I hope that the complicated processing of the variables used will be simplified by an adequate computerized program that will enable the automated calculation of the predicted risk.

PROGRESS-CKD is meant to be embedded and integrated in Health Information Systems. Therefore, the calculation is automatically triggered by entering new relevant information into the electronic health record of each patient. Currently PROGRES-CKD is embedded in the EuCliD®, a Health Information System which is available for use to all physicians and nurses working in Nephrocare clinics. Starting from 2022 a new version of EuCliD® called TheHub® will be made available to third-party clients in selected countries. Rollout of TheHub® worldwide will continue in a phased distribution schedule. It is important to remark that score calculation is based on demographic and laboratory variables that are typically accessible in real-world clinical practice: PROGRES-CKD automatically queries the information from the Health Information Systems, perform the calculation and returns the risk metrics in a graphical dashboard without any additional effort by the physician.

With regard to the output of PROGRES-CKD, results are presented in a simple graphical dashboard and are easily interpretable (please see Supplementary figure 1). In fact, in addition to provide an absolute value of "Patient's Risk", the scores are presented according to 5 categories: 1) <5% (very low risk), 2) 5-10% (low risk), 3) 10-20% (average risk), 4) 20-40% (high risk), 5) >40% (very high risk). The temporal dynamics of patients’ risk is also graphically represented in a line plot and key metrics (value of importance and risk factors importance) are graphically represented by color-coding diagrams.

  1. The case presented in the non-published material shows that the patient has a low risk of CKD progression during 24 months. It would be important to add (the methodology) which level of risk shows that the CKD progression will be exactly in 6 or 24 months. 

PROGRES-CKD is not a threshold model, rather it provides CKD progression risk estimation and other valuable information which can help nephrologists in clinical decision making on kidney replacement therapy. In fact, PROGRES-CKD computes the absolute risk of disease progression into kidney failure (KF) within either 6 or 24 months. The score is continuous in nature, anchored at 0.00=no risk at all to 1.00=certainty of failure within the prediction horizon. To facilitate data interpretability, PROGRES-CKD classifies the estimated risk in categories and displayed them in a traffic light sign representing the level of risk in quintiles of the risk distribution.

In addition, PROGRES-CKD can enhance physicians’ prognostic reasoning, by providing patient-specific impact metrics which can be used to prioritize further diagnostic testing as well as to estimate the potential impact of tailored interventions targeting modifiable risk factors.

Reviewer 2 Report

I would like to start by congratulating the authors and congratulating them for having decided to investigate an area where there is still so much to discover, but also for having decided to share this findings with the rest of the scientific community, so that science can evolve.

This is an article about a validation of a novel predictive algorithm for kidney failure in patients suffering from chronic kidney disease.

All comments, questions and suggestions presented are constructive and try to improve the article, after several careful readings.

Abstract

Start of sentence must be capitalized. Check if the sentences are well constructed. Check word spacing and punctuation.

Keywords

Repetitions with expressions that are in the title should be avoided. Whenever possible, keywords should be Mesh.

Introduction

Check word spacing and punctuation.

Material and methods

Figures should have more quality.

Results

Figures should have more quality.

Graphics should have more quality.

Discussion and Conclusions

Figure 5 should appear as close as possible to its reference.

General comments

Very interesting article, with clear potential to be published and very well written.

This article has the possibility of changing the clinical practice of nephrologists all over the world and of changing the way in which clinical decisions are made.

The article as presented must submitted to major revision, especially with regard to the quality of the images/graphics presented.

Author Response

Reviewer #2

  1. Abstract: Start of sentence must be capitalized. Check if the sentences are well constructed. Check word spacing and punctuation.

We thank the Reviewer for pointing this out. After a thorough review of the Abstract section, these inaccuracies have been corrected.

  1. Keywords: Repetitions with expressions that are in the title should be avoided. Whenever possible, keywords should be Mesh.

Accepting the Reviewer’s suggestion, the keyword “Kidney Failure (KF)” has been replaced with “End-stage kidney disease (ESKD)”.

We confirm all the keywords are Mesh terms, except for “Risk prediction” and “Naïve-Bayes Classifiers”.

  1. Introduction: Check word spacing and punctuation.

Thank you, we checked the Introduction and fixed all the mistakes.

  1. Material and methods: Figures should have more quality.
  2. Results: Figures should have more quality. Graphics should have more quality.
  3. Discussion and Conclusions: Figure 5 should appear as close as possible to its reference.

All the Figures and Graphics have been removed from the Manuscript file and will be separately uploaded as higher resolution images (.tif).

Round 2

Reviewer 2 Report

The authors carried out an important review of their article. My opinion is that it should be accepted for publication.